# Optimising Multimodality Treatment of Resectable Oesophago-Gastric Adenocarcinoma

**DOI:** 10.3390/cancers14030586

**Published:** 2022-01-24

**Authors:** Ali Abdulnabi Suwaidan, Anderley Gordon, Elizabeth Cartwright, David Cunningham

**Affiliations:** The Royal Marsden NHS Foundation Trust, London SW3 6JJ, UK; ali.suwaidan@nhs.net (A.A.S.); anderley.gordon@rmh.nhs.uk (A.G.); elizabeth.cartwright@rmh.nhs.uk (E.C.)

**Keywords:** gastric cancer, oesophageal cancer, multimodality treatment, precision oncology, immunotherapy

## Abstract

**Simple Summary:**

Oesophageal (food pipe) and stomach cancers are amongst the hard-to-treat cancers that result in significant illness and deaths around the globe. Over the last few decades, there has been remarkable progress in the treatment of these cancers as a result of advances in diagnosis, surgical techniques, systemic therapy and radiotherapy. However, even if caught in the early stages, most patients with these cancers will unfortunately have their cancers come back, usually becoming widespread and difficult to treat. Therefore, optimising the early treatment strategy of these cancers is essential to improve the outcome and reduce the risk of recurrence. There are currently various geographically influenced standard of care management practices of early stomach and oesophageal cancers, ranging from using chemotherapy before and after surgery to the use of combined chemoradiotherapy before surgery and more recently the use of immunotherapy after surgery. However, it is not very clear if one strategy is significantly better than the others and there are some ongoing studies aiming to directly compare these treatment options. In addition, our understanding of the molecular and genetic features of these cancers can help improve our clinical practice and inform our choice of the best treatment strategy for the individual patient.

**Abstract:**

Oesophago–gastric adenocarcinoma remains a leading cause of cancer-related morbidity and mortality worldwide. Although there has been an enormous progress in the multimodality management of resectable oesophago–gastric adenocarcinoma, most patients still develop a recurrent disease that eventually becomes resistant to systemic therapy. Currently, there is no global consensus on the optimal multimodality approach and there are variations in accepted standards of care, ranging from preoperative chemoradiation to perioperative chemotherapy and, more recently, adjuvant immune checkpoint inhibitors. Ongoing clinical trials are aimed to directly compare multimodal treatment options as well as the additional benefit of targeted therapies and immunotherapies. Furthermore, our understanding of the molecular and genetic features of oesophago–gastric cancer has improved significantly over the last decade and these data may help inform the best approach for the individual patient, utilising biomarker selection and precision medicine.

## 1. Introduction

Oesophago–gastric (OG) adenocarcinoma is a leading cause of cancer mortality globally, with over a million deaths per year worldwide [1]. Whilst the incidence trend for gastric adenocarcinoma has been largely declining, there has been a steep increase in the incidence of oesophageal and junctional adenocarcinoma, particularly in high-income countries in parallel with increasing rates of obesity, Barrett’s oesophagus and gastro–oesophageal reflux disease [2,3]. Furthermore, the incidence of these cancers shows clear geographical variations with the highest rates of gastric adenocarcinoma observed in Eastern Asia and Eastern Europe while the highest rates of oesophageal adenocarcinoma are seen in Northern Europe and Northern America [2]. These variations have significant implications for informing national policymaking and cancer control strategies.

Approximately 20% of patients with OG cancers present with resectable disease [4]. Gastric and oesophageal cancers are staged according to the American Joint Committee on Cancer (AJCC)/Union for International Cancer Control (UICC) TNM staging system (8th edition). While very early stage OG adenocarcinoma can be treated and cured with endoscopic resection or surgery alone, locally advanced disease requires multimodality treatment which has now become standard of care [5,6] based on strong evidence that established the role of perioperative chemotherapy and pre- and postoperative chemoradiotherapy. However, there is currently no global consensus on the optimal multimodality strategy to be advocated.

There are regional and geographical variations in the multimodality strategies ranging from the use of the perioperative chemotherapy strategy in Europe and Australasia, adjuvant chemotherapy in East Asia to the utilisation of pre- or postoperative chemoradiation in North America.

In this article, we discuss the current management strategies for resectable OG adenocarcinoma, the evidence from clinical studies, the biomarkers of response and future directions for further optimising and personalising the multimodality strategy.

## 2. Multimodality Strategies

### 2.1. Perioperative and Neoadjuvant Chemotherapy

Two large phase III randomised controlled trials established the practice of perioperative chemotherapy in patients with resectable OG adenocarcinomas: the UK Medical Research Council Adjuvant Gastric Infusional Chemotherapy (MAGIC) study and the Fédération Nationale des Centres de Lutte contre le Cancer (FNCLCC)/the Fédération Francophone de Cancérologie Digestive (FFCD) study. The main perioperative and neoadjuvant studies are summarised in Table 1.

In the pivotal MAGIC trial, 503 patients with resectable gastric (74%), gastro–oesophageal junction (GOJ, 14.5%) and lower oesophageal (11.5%) adenocarcinoma were randomly allocated to either perioperative chemotherapy consisting of three pre- and three post-operative cycles of epirubicin, cisplatin and 5-flourouracil (ECF) and surgery or to surgery alone. Compared to the surgery-only arm, patients in the perioperative chemotherapy arm had a significantly improved overall survival (OS) (hazard ratio (HR) 0.75, 95% CI 0.60–0.93; *p* = 0.009) and progression-free survival (HR 0.66, 95% CI 0.53–0.81; *p* < 0.001) with similar post-operative morbidity and mortality in both arms [8]. The 5-year OS rates were also significantly higher in the chemotherapy arm (36% vs. 23%; *p* = 0.009). Whilst 86% (N = 215/250) of patients assigned to the chemotherapy arm completed all preoperative chemotherapy, only 50% (N = 104/209) completed the three postoperative cycles, emphasizing the challenge of delivering adjuvant chemotherapy after a major surgery. This trial provided a paradigm shift in the standard of care treatment of locally advanced OG adenocarcinoma [9].

The perioperative strategy was reinforced in the French FNCLCC-FFCD phase III trial that randomised 224 patients with resectable adenocarcinoma of the lower oesophagus (11%), GOJ (64%) and stomach (25%) to either perioperative cisplatin-5-FU (CF) chemotherapy plus surgery or surgery alone. Analysis showed the perioperative chemotherapy group had higher OS (HR 0.69, 95% CI 0.50–0.95; *p* = 0.02) and disease-free survival (DFS) (HR 0.65, 95% CI 0.48–0.89; *p* = 0.003) [10].

In both trials, perioperative chemotherapy was associated with increased negative margin (R0) resection rates and favourable pathological tumour and nodal stage at resection compared to surgery alone [8,10]. As the gastric and GOJ tumours were the predominant anatomic subtypes in the MAGIC and FNCLCC-FFCD trials, respectively, these studies provided evidence for perioperative strategy in treating these tumours, particularly in Europe [6].

A large meta-analysis of 14 randomised controlled trials including the MAGIC and FNCLCC-FFCD trials showed perioperative chemotherapy in addition to surgery in resectable locally advanced OG adenocarcinoma was associated with longer OS (HR 0.81, 95% CI 0.73–0.89), longer disease-free survival, higher rates of R0 resection, and more favourable tumour stage upon resection, while there was no association with perioperative morbidity and mortality [11]. The absolute survival benefit at 5 years was 9%.

The practice-changing German phase III FLOT4 trial established a new perioperative regimen of triplet chemotherapy with 5-FU, leucovorin, oxaliplatin and docetaxel (FLOT) as a superior alternative to the MAGIC regimen (ECF/ECX) [12]. In this study, 716 patients with cT2 or higher and/or nodal positive stage (N+) gastric or GOJ tumours were randomised to receive either four 2-weekly cycles of pre- and postoperative FLOT chemotherapy or three 3-weekly cycles of perioperative ECF/ECX. In the intention-to-treat population, more patients achieved R0 resection in the FLOT group than in the ECF/ECX group (85% vs. 78%) [12] with higher proportion of complete pathological regression (16% vs. 6%) demonstrated in the preceding phase II of the same study [13]. Perioperative FLOT was associated with a significant improvement in the median OS from 35 to 50 months (HR 0.77; CI, 0.63–0.94; *p* = 0.012) with a projected 5-year OS of 45% vs. 36% for perioperative ECF/ECX [12]. Of note, more patients in the FLOT group commenced postoperative chemotherapy (60%) compared to ECF/ECX group (52%) and therefore more patients completed treatment protocol (46% vs. 37%). Although FLOT was associated with higher rates of grade 3–4 infections, neutropenia, diarrhoea and neuropathy, the rates of post-operative complications and hospitalisation for toxicity were similar in both groups. Based on the results of this trial, FLOT is now the recommended standard of care in perioperative chemotherapy for patients who are candidates for triplet chemotherapy, and it has been adopted in the European and American clinical practice guidelines.

A purely preoperative chemotherapy approach without adjuvant chemotherapy was evaluated in the EORTC 40954 Trial [14]. This study terminated early due to poor accrual (N = 144) and was underpowered for OS assessment. Two cycles of preoperative cisplatin and 5-FU was associated with higher rates of R0 resection (81.9% vs. 66.7%, *p* = 0.036), however it did not result in any significant survival benefit compared to surgery alone in patients with gastric and GOJ tumours (HR 0.84, 95% CI 0.52–1.35; *p* = 0.466). The authors of the study discussed potential reasons for these results beside the omission of adjuvant chemotherapy. They commented on the higher rates of D2 lymphadenectomy (95%) compared to around 40% in the MAGIC trial, which may have marginalised the contribution of neoadjuvant chemotherapy. In addition, the rate of completion of preoperative chemotherapy was lower in this trial compared to the MAGIC and FNCLCC-FFCD studies (65% vs. 86% and 87%, respectively). 

In contrast, the MRC OE02 trial reported significant improvement in the 5-year OS (23% vs.17.1%, HR 0.84; 95% CI 0.72–0.98; *p* = 0.03) with two cycles of preoperative chemotherapy (CF) compared with surgery alone in patients with oesophageal adeno- and squamous cell carcinoma [15]. Furthermore, two subsequent meta-analyses including 1724 patients [16] and 2102 patients [17] have confirmed the survival advantage for preoperative chemotherapy in resectable oesophageal adenocarcinoma, with a 2-year absolute survival benefit of 7% and 5-year absolute benefit of 4%, respectively, with most benefit being seen among patients with junctional tumours and those with adenocarcinomas [16].

The MRC OE05 trial showed that intensification of preoperative chemotherapy with four cycles of epirubicin, cisplatin and capecitabin (ECX) in resectable oesophageal and GOJ (Siewert type 1 and 2) did not improve median OS compared to two cycles of CF (median OS 26.1 vs. 23.4 months, HR 0.90, 95% CI 0.77–1.05; *p* = 0.19) [18]. 

The attributable efficacy of neoadjuvant or adjuvant chemotherapy is unclear and only few trials directly compared both approaches. The SAKK 43/99 randomised 69 patients to either neoadjuvant or adjuvant docetaxel-based triplet chemotherapy with cisplatin and 5-FU (TCF). The trial discontinued early due to slow accrual and was thus underpowered for survival analysis [19]. Downstaging was achieved in 19/32 (60%) patients who received preoperative TCF chemotherapy, 4 patients (18%) having a complete pathological response (pCR) and 18 patients (53%) a partial microscopic pathological response (pPR) [20]. The 10-year follow-up results did not reveal any significant difference in event-free survival and OS [20].

### 2.2. Adjuvant Chemotherapy 

The practice of adjuvant chemotherapy has been largely employed in East Asia, where treatment paradigms are shaped by high gastric cancer incidence, national screening programmes and an early stage at diagnosis [21]. Two pivotal adjuvant trials have established this practice in Asian populations (Table 2). The Japanese Adjuvant Chemotherapy Trial of S-1 for Gastric Cancer (ACTS-GC) trial investigated the role of adjuvant S-1 (oral fluoropyrimidine) postoperatively for 12 months following D2 surgery for stage II and III gastric adenocarcinoma [22]. The trial was stopped early (N = 1069) at interim analysis, showing that the S-1 group had a higher OS rate compared to the surgery-only group (*p* = 0.002). The 3-year OS rate was 10% higher in the adjuvant S-1 group (80.1% vs. 70.1%, HR 0.68, 95% CI 0.52–0.87, *p* = 0.003). The updated 5-year outcomes showed a maintained survival benefit (71.7% vs. 61.1%, HR 0.669, 95% CI 0.540–0.828) [23]. The most common grade 3–4 adverse events in the S-1 group were anorexia (6%), nausea (3.7%) and diarrhoea (3.1%). Based on this trial, adjuvant S-1 became the standard practice in East Asia.

The non-inferiority JCOG1104 trial sought to evaluate whether four cycles (6 months) of adjuvant S-1 was non-inferior to eight cycles (12 months) in stage II gastric cancer. However, this study was terminated early because the point estimate of the HR for recurrence at the first planned interim analysis (HR 2.52, 95% CI 1.11–5.77) was greater than the non-inferiority margin of the HR (1.37), a prespecified criterion for early stopping. The authors concluded that S-1 for 1 year should remain the standard duration of adjuvant chemotherapy [24].

More recently, the phase III JACCRO GC-07 (START-2) trial demonstrated that the combination of S-1 with docetaxel (six cycles) was superior to S-1 alone as adjuvant therapy after D2 resection in stage III gastric cancer. The 3-year relapse-free survival (RFS) rate was 67.7% in the S-1 plus docetaxel group compared to 57.4% in the S-1 group (HR 0.715, 95% CI 0.587–0.871; *p* = 0.0008). This also translated into a significant benefit in the 3-year OS rate (77.7% vs. 71.2%, HR 0.742, 95% CI 0.596–0.925; *p* = 0.0076) [25].

The phase III CLASSIC trial evaluated the efficacy of oxaliplatin-based adjuvant chemotherapy in stage II and III gastric cancers after D2 resection. A total of 1035 patients from 35 centres in China, South Korea, and Taiwan were randomly assigned to either receive eight cycles of adjuvant capecitabine and oxaliplatin for 6 months or observation only. Patients who received adjuvant chemotherapy had improved DFS (HR 0.58, 95% CI 0.44–0.72; *p* < 0.0001) and OS (HR 0.66, 95% CI 0.51–0.85; *p*= 0.0015) compared to surgery and observation alone. The estimated 5-year OS rate was 78% (95% CI 74–82) in the adjuvant capecitabine and oxaliplatin group versus 69% (95% CI 64–73) in the observation group [26,27].

In a retrospective observational study of 1088 patients in Korea, adjuvant S-1 was associated with higher recurrence and lower OS rates compared to adjuvant capecitabine and oxaliplatin for patients with stage IIIB and IIIC gastric cancer after D2 resection [28]. No difference was found in stage II and IIIA disease. A randomised controlled trial comparing adjuvant S-1 and S-1 plus oxaliplatin is currently recruiting in China (NCT02867839). 

Several studies explored adjuvant intensification with sequential chemotherapies. In a two-by-two factorial trial design, the Japanese SAMIT trial randomly assigned 1495 patients with T4a or T4b gastric cancer after D2 gastrectomy to one of four treatment groups: S-1 only, UFT (tegafur, uracil) only, or paclitaxel followed by either S-1 or UFT. The paclitaxel added sequentially to adjuvant S-1 or UFT did not improve survival (3-year DFS 57.2% vs. 54.0%, HR 0.92, 95% CI 0.80–1.07; *p* = 0.273) and UFT was not non-inferior to S-1 in the 3-year DFS (53% vs. 58.2%, HR 0.81, 95% CI 0.70–0.93; *p* = 0.0048) [29]. Similarly in Europe, the Italian ITACA-S showed that intensification of adjuvant chemotherapy by sequencing three cycles of docetaxel plus cisplatin after four cycles of FOLIRI failed to improve DFS or OS as compared to nine cycles of 5-FU/leucovorin only [30].

A large individual patient-level meta-analysis of adjuvant chemotherapy in gastric cancer has confirmed a 5.7% absolute benefit for fluorouracil (5-FU)-based chemotherapy, compared with surgery alone (HR 0.82, 95% CI 0.76–0.90; *p* < 0.001) in all regimen groups tested (monochemotherapy; fluorouracil and mitomycin C with anthracyclines; fluorouracil, mitomycin C, and others without anthracyclines; other polychemotherapy) [31].

Although the evidence for adjuvant chemotherapy was mainly based in Asian populations, the European clinical practice guidelines recommend adjuvant chemotherapy or chemoradiotherapy for patients with gastric cancer who underwent surgery without preoperative chemotherapy (e.g., due to understaging before the initial decision for upfront surgery) [6]. For oesophageal adenocarcinoma, there is a paucity of evidence to support adjuvant chemotherapy alone.

### 2.3. Adjuvant or Neoadjuvant Chemoradiotherapy

The role of chemoradiotherapy has been investigated in both the pre- and postoperative settings of OG cancers in several studies (Table 3).

The American phase III Intergroup-0116 trial randomly assigned 556 patients with gastric or GOJ adenocarcinoma to either surgery plus postoperative chemoradiotherapy (five monthly cycles of bolus 5-FU and leucovorin for 5 days with concurrent radiotherapy 45 Gy in 25 fractions over 5 weeks with cycles two and three with lower dose 5-FU) or surgery alone. With a median follow-up period of 5 years, a 9-month improvement in the median OS was detected in the chemoradiotherapy arm of the study (27 vs. 36 months, HR 1.35, 95% CI 1.09–1.66; *p* = 0.005) and an 11-month improvement in the median RFS (19 vs. 30 months, HR 1.52, 95% CI 1.23–1.86; *p* < 0.001). This was mainly driven by reduced local and regional recurrence in the chemoradiotherapy group (19% vs. 29% and 65% vs. 72%, respectively) [32]. However, this trial was criticised for the extent of lymphadenectomy, as 54% of patients had D0 resection and only 10% had a D2 resection. This was argued to be an inadequate surgical resection and that the study merely illustrated the benefit of chemoradiotherapy to compensate for the suboptimal surgery [33,34]. However, a post hoc analysis by the study authors showed no difference in the treatment outcome by the level of surgery performed [35] and the updated analysis of this trial after more than 10-year median follow up showed continued strong benefit from the tri-modality treatment for both OS (HR = 1.32, 95% CI 1.10–1.60; *p* = 0.0046) and DFS (HR = 1.51, 95% CI 1.25–1.83; *p* < 0.001) [36]. Based on the results of this trial, adjuvant chemoradiotherapy has been widely adopted as a standard adjunctive therapy after curative surgery for OG cancers in North America [37].

In contrast, the Korean phase III ARTIST trial showed no significant improvement in the DFS with the addition of radiotherapy to adjuvant capecitabine and cisplatin (XP) chemotherapy after D2 gastrectomy (3-year DFS 78.2% vs. 74.2%, *p* = 0.0862) [38] which remained similar after 7 years of follow up (HR 0.74, 95% CI 0.520–1.050; *p* = 0.0922) [39]. Although subgroup analysis in this trial showed that adjuvant chemoradiotherapy significantly improved DFS in patients with node-positive disease (77.5% vs.72.3%, *p* = 0.0365) [38], a subsequent trial (ARTIST-II) specifically designed to assess this subgroup of patients (D2-resected, node-positive gastric cancer) recently reported no significant difference in DFS with the addition of radiotherapy to adjuvant S1 and oxaliplatin (SOX) chemotherapy [40]. Similarly, the CRITICS trial based in Netherlands, Sweden and Denmark showed no significant difference in the median OS with adjuvant chemoradiotherapy (CX and 45 Gy in 25 fractions) compared to adjuvant ECF/ECX after preoperative ECF/ECX and surgery (37 vs. 43 months, HR 1.01, 95% CI 0.84–1.22; *p* = 0.90) in stage IB–IVA resectable gastric or GOJ adenocarcinoma [41].

The benefit of neoadjuvant chemoradiation has been established for OG cancers in two randomised trials. The Dutch CROSS trial randomized 368 patients with non-metastatic T1N1 or T2-3N0-1 oesophageal or GOJ adenocarcinoma (75%) or squamous cell carcinoma (23%) to either preoperative chemoradiotherapy (weekly carboplatin and paclitaxel with concurrent 41.4 Gy in 23 fractions over 5 weeks) followed by surgery or surgery alone. Compared to surgery alone, preoperative chemotherapy resulted in higher rates of complete resection (R0) (92% vs. 69%, *p* < 0.001), reduced rates of lymph node involvement (31% vs. 75%, *p* < 0.001) and improved OS (median OS 49.4 vs. 24.0 months, HR 0.657, 95% CI, 0.495 to 0.871: *p* = 0.003) [42]. However, pathological complete response rate was less pronounced in the adenocarcinoma subgroup compared to squamous cell carcinoma (23% vs. 49%, *p* = 0.008, respectively). 

For gastric adenocarcinoma, the phase II RTOG 9904 demonstrated high rates of pathological responses and R0 resection with preoperative induction chemotherapy (two cycles of 5-FU, leucovorin and cisplatin) followed by radiotherapy with weekly 5-FU and paclitaxel (26% and 77%, respectively) [43] which compares favourably to the CROSS effect in oesophageal and GOJ adenocarcinoma. Preoperative chemotherapy or chemoradiotherapy has distinct biological and clinical advantages compared with postoperative therapy, including tumour downstaging with an increase in R0 resection, and better patient tolerability [44].

Preliminary results of the phase III randomised control trial Neo-AEGIS (NCT01726452) comparing neoadjuvant CROSS (carboplatin/paclitaxel, 41.4Gy radiation) and optimum perioperative chemotherapy regimens, including modified MAGIC ECF/X and FLOT, in patients with T2-3, N0-3 locally advanced adenocarcinoma of the oesophagus and GOJ were reported at the American Society of Clinical Oncology Annual Meeting 2021 [45]. A total of 377 patients were randomly assigned in a 1:1 fashion to CROSS or perioperative chemotherapy (either epirubicin, cisplatin (oxaliplatin), and 5-FU (capecitabine) pre-2018 or FLOT from 2019/20) at 24 sites in five countries (Ireland, UK, Denmark, France and Sweden). Of the 362 evaluable patients, there were 143 deaths, 70 in CROSS arm and 73 in the MAGIC/FLOT arm, with 3-year estimated survival probability of 56% (95% CI 47–64) and 57% (95% CI 48–65), respectively. These preliminary results show no evidence that perioperative chemotherapy is unacceptably inferior to neoadjuvant chemoradiation and support either treatment approach [45]. However, most of the enrolled patients in the perioperative chemotherapy arm of this trial have not received FLOT regimen (27 of 184 only) which is the current standard of care. Therefore, the multicenter RACE trial (NCT04375605) is ongoing to evaluate the benefit of adding radiotherapy to FLOT chemotherapy in patients with GOJ adenocarcinomas [46].

### 2.4. The Role of Immune Checkpoint Inhibitors

Immune checkpoint inhibitors, mainly anti-PD1/anti-PD-L1, have shown promising results in the advanced/metastatic setting of OG cancers. In early-stage disease, there is no established adjuvant treatment after neoadjuvant CRT. The phase III CheckMate 577 trial (N = 794) sought to evaluate the efficacy of adjuvant nivolumab, an anti-PD1 monoclonal antibody, in patients with R0 resected stage II/III oesophageal and GOJ cancers (71% adenocarcinoma and 29% squamous cell carcinoma) with residual pathological disease and high risk of recurrence following neoadjuvant chemoradiotherapy. Patients were randomised in a 2:1 fashion to either adjuvant nivolumab 240 mg or placebo, twice-weekly for 16 weeks. At pre-specified interim analysis, compared to placebo, adjuvant nivolumab led to a significant improvement in DFS (22.4 vs. 11.0 months, HR 0.69, 96.4% CI 0.56–0.86; *p* < 0.0003) [47] representing a new standard of care option. In a post hoc analysis, the DFS benefit was seen in tumours with a PD-L1 combined positive score (CPS) of <5 as well as ≥5, with a hazard ratio of (0.62 [95% CI 0.46–0.83] vs. 0.89 [0.65–1.22]) respectively [47].

In Asia, where adjuvant chemotherapy is standard of care in patients with resectable gastric and junctional cancers, ATTRACTION-05 (NCT03006705) is currently evaluating combination chemoimmunotherapy as adjuvant treatment. In this double-blind placebo-controlled phase III study, patients with pathological stage III gastric and gastro–oesophageal junction cancer after D2 lymphadenectomy will be randomised to either investigator’s choice S-1 or CAPOX in combination with nivolumab or placebo. In the perioperative setting, the EORTC VESTIGE trial (NCT03006705) is a phase II study seeking to evaluate the role of immunotherapy as an alternative to continuing with post-operative chemotherapy with the same preoperative regimen, as is currently standard of care with a perioperative approach, in patients with high risk gastric and oesophageal adenocarcinoma. In this study, patients with node positive and/or R1 resection following neoadjuvant chemotherapy and surgery are randomised to receive either adjuvant chemotherapy (same regimen as pre-operation) or nivolumab (3 mg/kg IV 2-weekly) and ipilimumab 1 mg/kg IV 6-weekly). 

Several FLOT-immunotherapy combination studies are ongoing in the perioperative setting, KEYNOTE 585 (NCT03221426—FLOT in combination with pembrolizumab or placebo), DANTE/FLOT8 (NCT0341288—FLOT in combination with atezolizumab), ICONIC (NCT03399071—FLOT in combination with avelumab) and efficacy results are awaited. A Japanese open-label phase II study (NCT04745988) of pembrolizumab in combination with lenvatinib, a multi-kinase VEGF inhibitor delivered for 3 cycles pre- and postsurgery followed by 11 cycles (thrice-weekly) of pembrolizumab monotherapy in patients with T2-4 and/or node positive localised gastric and gastro–oesophageal cancers is seeking to evaluate the safety and efficacy of a chemotherapy-free perioperative approach with major pathological response as the primary endpoint. Active studies investigating the role of chemoimmunotherapy combination in the perioperative studies are summarised in Table 4.

### 2.5. Perioperative Targeted Therapy

The addition of targeted therapies (anti-VEGF and anti-HER2 monoclonal antibodies) to chemotherapy backbone has been established as standard of care in the treatment of unresectable/metastatic OG adenocarcinoma [48,49,50,51]. This has encouraged investigating the role of these targeted therapies in the perioperative setting. However, the MRC ST03 trial did not show any survival advantage from the addition of bevacizumab, an anti-VEGFR monoclonal antibody, to perioperative ECX chemotherapy (3-year OS was 48.1% vs. 50.3%, HR 1.08, 95% CI 0.91–1.29; *p* = 0.36). Furthermore, the rates of anastomotic leak and wound infections postoperatively were higher in the bevacizumab arm [52]. The phase II FLOT7-RAMSES trial reported recently that the addition of ramucirumab, an anti-VEGFR2 monoclonal antibody, to perioperative FLOT improved R0 resection rates (97% vs. 83%, *p* = 0.0049) with no impact on pathological response (27% vs. 30%, *p* = 0.7363) [53]. There was a higher mortality rate with FLOT plus ramucirumab compared to FLOT alone (5.9% vs. 2.5%); however, mortality rate was similar in both groups (2.9%) after excluding type I GOJ adenocarcinoma from analysis [53]. The phase III part of this study is ongoing after excluding type 1 GOJ adenocarcinoma.

In a small Spanish phase II trial, the feasibility of adding trastuzumab, an anti-HER2 monoclonal antibody, to perioperative chemotherapy with capecitabine and oxaliplatin was determined in 36 patients with resectable HER2 positive gastric or GOJ adenocarcinoma, where the 18-month DFS was 71% [54]. The PETRARCA trial reported that the addition of trastuzumab and pertuzumab to perioperative FLOT chemotherapy significantly improved the rates of pathological complete response (35% vs. 12%, *p* = 0.02) and pathological nodal negativity rates (68% vs. 39%) in HER2 positive resectable OG adenocarcinoma. This was, however, at the expense of higher rates of diarrhoea (41% vs. 5%) and leukopenia (23% vs. 13%) [55]. This trial closed prematurely and did not proceed to phase III.

The recently completed Dutch HER-FLOT trial investigated the combination of trastuzumab with FLOT in 56 patients. In this single-arm phase II trial (N = 56), 12/56 (21.4%) patients with locally advanced HER2 positive (2+ or 3+) OG adenocarcinoma had complete pathological response following neoadjuvant FLOT and trastuzumab, meeting the primary endpoint of the study (pre-specified threshold pCR > 20%) [56]. A further 14/56 (25%) patients had near complete response with a three-year OS of 82.1% (95% CI 69.1–91.1). The phase II EORTC INNOVATION trial is currently investigating combination HER2 targeting with 1:2:2 randomisation to chemotherapy alone, chemotherapy plus trastuzumab, or chemotherapy plus dual HER2 blockade [57].

In combination with chemoradiation, the RTOG 1010 study (N = 203) showed no improvement in DFS with the addition of trastuzumab to trimodality neoadjuvant chemoradiotherapy (weekly pacliataxel and carboplatin plus 50.4 Gy in 28 fractions over 6 weeks) in resectable HER2 positive oesophageal adenocarcinoma (HR 0.97 0.69–1.36, *p* = 0.85) [58].

## 3. Biomarkers in Resectable OG Adenocarcinoma

Several tissue and circulating biomarkers have been explored in resectable OG cancer through landmark clinical trials over recent years. The Cancer Genome Atlas (TGCA) [59] identified four molecular subtypes of OG cancer—EBV-positive, microsatellite instability (MSI), genomic stable and chromosomal unstable tumour subtypes—and the Asian Cancer Research Group (ACRG) identified MSI, microsatellite stable/epithelial to mesenchymal transition (MSS/EMT), MSS/TP53+ and MSS/TP53− as subtypes with prognostic values [60]. These classification systems may serve as a valuable adjunct to histopathology to further enable precision medicine in OG cancer through understanding and targeting the genomic features that may drive tumorigenesis [59]. It is important to note that some of these biomarkers, particularly MSI-high and EBV-positive status, have shown promising roles in predicting response to immune checkpoint inhibitors in the advanced unresectable and metastatic gastric cancer [61,62].

### 3.1. MSI and Mismatch Repair (MMR)

MSI and MMR status are two of the most studied molecular biomarkers in solid tumours. In resected gastric cancer, post hoc analysis from the MAGIC and CLASSIC trials demonstrated a relationship between MSI status and clinical outcome. In the CLASSIC trial, patients with MSI-high tumours (40/592) derived no survival benefit from adjuvant chemotherapy (5-year DFS for chemotherapy versus surgery alone: 83.9% vs. 85.7%; *p* = 0.931 [63]. In the MAGIC trial, MSI-high tumours were present in 20 of the 303 and had better survival in the surgery alone arm compared to MSI-low or microsatellite stable (MSS) tumours (median OS not reached vs.20.3 months, HR 0.35, 95% CI, 0.11–1.11; *p*  =  0.08). Concordance between MSI-high and MMR deficient (dMMR) status was 97.6%, therefore dMMR tumours also demonstrated superior survival in patients undergoing surgery alone compared to MMR proficient tumours. In the dMMR and MSI-high patients who received chemotherapy plus surgery, the median OS was 9.6 months (95% CI, 0.1–22.5 months) compared 19.5 months in the MSI-low/stable and MMR proficient patients (95% CI, 15.4–35.2 months; HR 2.18, 95% CI 1.08–4.42; *p*  =  0.03) [64]. These results were significant and aligned with other Asian studies [65,66,67,68]. However, the prevalence of MSI-high patients were low in both studies therefore Pietrantonio and colleagues performed a multinational individual patient data meta-analysis of 1556 patients incorporating data from the MAGIC, CLASSIC, ARTIST and ITACA-S trials to provide a more robust assessment of the MSI status in OG cancers [69]. This analysis supports the role of MSI as a positive prognostic biomarker in resected gastric cancer and it suggested that MSI-high patients may not derive any benefit from perioperative or adjuvant chemotherapy compared to surgery alone (5-year DFS: 70% versus 77% respectively (HR, 1.27; 95% CI 0.53 to 3.04), and 5-year OS 75% versus 83% (HR, 1.50; 95% CI 0.55 to 4.12)). According to the TCGA molecular classification, 22% of gastric cancers were MSI-high and these tumours were associated with elevated mutation rates, including mutations of genes encoding targetable oncogenic signalling proteins [59]. This feature may suggest these tumours to be more sensitive to immunotherapy; however, within resectable OG cancer this is an area of ongoing investigation (NCT04006262, NCT04795661, NCT04152889). MSI/MMR status has a considerable potential to be utilized clinically as it could improve patient selection in the adjuvant setting and identify patients who may derive a benefit from immunotherapy. Nonetheless, validation is needed through prospective large randomized clinical trials.

### 3.2. Programmed Death Ligand-1 (PDL-1) 

PD-L1 upregulation is present in more than 40% of OG cancers [70]. Eto and colleagues showed in a study of 105 Japanese patients who underwent curative gastrectomy for stage II/III gastric cancer that PD-1 expression was associated with poorer DFS (3 y DFS 36.1% vs. 64.7% respectively, *p* < 0.05) as well as poorer OS [71]. This was also demonstrated in other Asian studies [72,73]. In contrast, Böger and colleagues analysed 465 gastric cancer samples from Caucasian patients who had undergone total or partial gastrectomy for OG adenocarcinoma, as well as 15 corresponding liver metastases. PD-L1 positivity was evaluated using a simplified immunoreactivity scoring system (IRS), which incorporated the percentage of immunoreactive cells and intensity of immunostaining. Using IRS scoring, patients with PD-L1 positive tumour and immune cells had a significant better overall survival (*p* = 0.028) [74].

In a small phase II trial, 37 patients with locally advanced OG adenocarcinoma who had completed CRT and undergone R0 resection but had persistent residual disease in the surgical specimen were treated with adjuvant durvulumab for up to 1 year. This study showed a favourable trend toward superior 1-year RFS in patients with PD-L1 positive versus PD-L1 negative tumours. All of the 7 patients (18%) with tumours expressing PD-L1 CPS ≥ 10 were alive at 1 year. Of the 19 patients (51%) with PD-L1 CPS ≥ 1, median OS was significantly superior compared to patients with PD-L1 CPS < 1 (40.4 vs. 25.0 months, *p* = 0.0132) [75]. Though this study was underpowered due to small study size, it highlights the potential for PD-L1 to be used as a predictive biomarker to immunotherapy in the adjuvant setting for resected OG adenocarcinoma.

Within advanced OG adenocarcinoma, nivolumab has been FDA approved in the first line metastatic setting for OG adenocarcinoma based on results from CheckMate 649. The OS benefit was significant across all the PD-L1 CPS, although this was more pronounced in the CPS ≥ 5 [76]. The role of PD-L1 expression is certainly established in gastric cancer and its role in early stage OG adenocarcinoma is being investigated in trials outlined in Table 4, with some trials stratifying treatment based on this biomarker. Although emerging as a predictive biomarker of response to immunotherapy, a standardised cut off value to measure PD-L1 expression is required. 

A large retrospective study using >1600 gastric cancer samples showed distinct differences in the tumour immunity gene expression signatures related to T-cell function between Asian and non-Asian patients, which may have implications in the differences in clinical outcomes as well as the predicting the response to immunotherapy [77].

### 3.3. Epstein Barr Virus (EBV)

EBV is a known carcinogenic agent that encompassed approximately 9% of gastric cancers as demonstrated by the TCGA study, which contained mostly early-stage disease [59]. EBV positive tumours display a distinct tumorigenic profile which includes robust PD-L1 expression, amplifications of chromosomal region 9p24.1, that encode PD-1 ligands, JAK2 amplifications, higher prevalence of DNA hypermethylation, recurrent PIK3CA mutations and different patterns of immune infiltrates [59]. In addition, Derks and colleagues demonstrated that EBV-positive tumours correlated to high PD-L1 expression, and they utilized gene expression profiling from the TCGA samples which showed interferon-γ driven gene signatures were enriched in EBV positive tumours [78].

Sohn et al. used genomic data from a TCGA cohort (N= 262) to analyse the four molecular subtypes of gastric cancer to develop prediction models. They concluded that EBV subtype was associated with the best prognosis for both RFS (*p* = 0.006) and OS (*p* = 0.004) [79]. The tumour landscape of EBV suggest this subtype may be a good candidate for PD-L1 targeted treatment. The French IMHOTEP trial (NCT04795661) is an active phase II trial evaluating neoadjuvant pembrolizumab in patients with MSI/dMMR- or EBV-positive gastric cancer, so this may provide more prognostic and predictive data on the EBV subtypes in this setting.

### 3.4. Human Epidermal Growth Factor Receptor 2 (HER2)

Approximately 10–20% of resectable gastric cancers are HER2 positive. The MAGIC trial demonstrated that HER2 status was neither a clinically useful biomarker for patient selection in perioperative treatment nor a prognostic biomarker in early OG cancer [80]. In the metastatic setting, HER2 positivity (2+ or 3+) was associated with significantly improved survival with chemotherapy and trastuzumab versus chemotherapy alone (HR 0.65, 95% CI 0.51–0.83) as demonstrated in the TOGA trial [48]. In resectable gastric cancer, the impact of HER2 as a predictive and prognostic biomarker is an ongoing area of investigation.

### 3.5. Other Biomarkers

Clinicopathological factors such as tumour regression grading (TRG), lymph node status and resection margin status are post-operative measures which provide prognostic information that can support clinical decision making in the adjuvant setting. Pathological tumour response is prognostic of survival and is under investigation in current IO-chemotherapy trials. There are five key classification systems used globally to measure TRG [81]. The Mandard and Becker classification systems have been shown to provide the most favourable prognostic information and are widely used. The majority of pivotal perioperative trials have demonstrated that either Mandard tumour regression grade 1 (complete regression) or 2 (fibrosis with scattered tumour cells) or Becker TRG1a (no residual cells), 1b (<10% residual tumour cells) and 2 (10–50% residual tumour cells) predict better survival outcomes [8,27]. The MAGIC trial denoted better 5-year OS in patients with TRG 1-2 than TRG 3-5 (58% vs. 28.9%, respectively) and that lymph node metastases was the only independently predictive factor of overall survival in patients after neoadjuvant chemotherapy (HR, 3.36; 95% CI, 1.70 to 6.63; *p* < 0.001) [82] The phase II FLOT-4 trial used Becker’s classification and achieved higher proportion of patients having complete pathological response (pCR) with docetaxel-based treatment vs. epirubicin-based triplet respectively (16% vs. 6%). TRG has a prognostic benefit, however a standardization grading system could make this more robust.

The use of 18F-fluorodeoxyglucose (FDG) PET-CT has been explored as a prognostic and predictive marker of early response to neoadjuvant treatment. In the MUNICON trial, 110 patients with locally advanced junctional adenocarcinoma had FDGPET/CT performed 2 weeks after induction fluoropyrimidine/platinum-based chemotherapy [83]. Metabolic responders continued with neoadjuvant chemotherapy and the metabolic non-responders proceeded to surgery. Of 110 patients, 54 (49%) demonstrated an early metabolic response after 2 weeks, which was defined as a decrease of 35% or more in tumour glucose standard uptake values (SUV). The trial reported improved overall survival in the early metabolic responders compared to the non-metabolic responders (NR vs. 25.8 months HR 2.13, 95% CI 23.6–35.7, *p* = 0.015) which demonstrated the potential of PET guided management in these tumours. These results lead to the MUNICON II prospective trial, which explored the clinical outcome of metabolic non-responders having salvage neoadjuvant radiochemotherapy versus metabolic responders, who had neoadjuvant chemotherapy alone for 3 months followed by surgery as a measure to stratify neoadjuvant treatment. Of the 56 patients who had induction chemotherapy, 33 were metabolic responders and 23 non-responders. The primary endpoint of increased R0 resection rate in the metabolic responders was not met (*p* = 0.51), and 1-year overall survival was comparable between both groups; however, 2-year overall survival was favoured in the metabolic responders compared to the non-responders (71% ± 8% and 42% ± 11%, respectively, HR 1.9, *p* = 0.10) [84].

### 3.6. Future Prospective Biomarkers

Liquid biopsies are emerging in gastrointestinal cancers as they have the potential to map out the genetic profile of a tumour, thereby having substantial potential in early detection as well as detecting minimal residual disease. Circulating free DNA (cfDNA) exists in plasma, serum, and/or other bodily fluids. Levels can be 2–3-fold higher in patients with cancer compared to healthy patients; thus, it has the potential to be a biomarker of early detection [85]. Kim and colleagues evaluated plasma cfDNA to discriminate cancer in 30 gastric cancer patients and 34 healthy patients. cfDNA levels were higher in gastric cancer patients both before and 24 h after surgery, and levels were more substantial in the advanced patients (N = 14) [86]. Similarly, Qian et al. demonstrated quantitative cfDNA concentration levels were higher in gastric cancer (N = 124) than in benign disease (n = 64) and healthy controls (N = 92) (*p ≤* 0.05) [87]. cfDNA is promising in early detection, however further research is needed to better define sensitivity and specificity for clinical utility. 

In the post-operative setting, circulating tumour DNA (ctDNA) has potential to detect molecular residual disease (MRD). In a prospective cohort study by Yang et al., 46 patients with stage I to III gastric cancer who underwent curative surgery had surgical tissue and plasma samples analysed for ctDNA evaluation. All patients who had ctDNA detected immediately after surgery developed disease recurrence and the presence of ctDNA was associated with poorer DFS and OS (HR = 14.78, 95%CI, 7.991–61.29, *p*  <  0.0001 and HR = 7.664, 95% CI, 2.916–21.06, *p*  =  0.002, respectively), suggesting its value in MRD detection [88].

## 4. Conclusions

Multimodality treatment with chemotherapy, radiotherapy and surgery are mainstay for the management of resectable OG adenocarcinomas. Despite this approach, around half of patients with early-stage disease will relapse and the challenge remains to optimise treatment for such patients.

Encouraging results of CheckMate 577 have shown a significant improvement in DFS with the addition of immunotherapy to multimodality treatment in early-stage disease. Overall survival data is awaited, and this approach has not been compared to perioperative chemotherapy; nevertheless, for a selected group of patients with residual pathological disease following neoadjuvant treatment and who are fit enough to tolerate this approach, adjuvant nivolumab has recently been established as a new standard of care.

The final results of this study and others, investigating immunotherapy combination strategies in early-stage disease, are awaited with interest. Delivering precision medicine in early-stage disease is one strategy for improving outcomes. In patients with HER2-positive disease, HER2 targeting in combination with chemotherapy has shown promising pathological complete response rates, and survival data is awaited.

In addition to optimising perioperative therapeutic strategies, ctDNA detection of minimal residual disease holds promise for identifying those patients who may benefit most from adjuvant therapy and identifying early disease relapse following curative intent treatment. Trials are ongoing to further evaluate the role of ctDNA as a predictive and prognostic biomarker with the future potential to inform the early-stage treatment paradigm.

## Figures and Tables

**Table 1 cancers-14-00586-t001:** Key studies of perioperative and neoadjuvant chemotherapy for resectable OG cancers.

Trial	Therapy Arms	N	Tumour Location	DFS, HR (95% CI)	OS, HR (95% CI)
MAGIC	Surgery alone vs. Perioperative ECF	253 250	Oesophageal 14.5% GOJ 11.5% Gastric 74%	HR 0.66 (0.53–0.81) *p* < 0.001	5-year OS 23% vs. 36% HR 0.75 (0.60–0.93) *p* = 0.009
FNCLCC/ FFCD	Surgery alone vs. Perioperative CF	111 113	Oesophageal 11% GOJ 64% Gastric 25%	5-year DFS 19% vs. 34% HR 0.65 (0.48–0.89) *p* = 0.003	5-year OS 24% vs. 38% HR 0.69 (0.50–0.95) *p* = 0.02
AIO-FLOT4	Perioperative ECF/ECX vs. Perioperative FLOT	360 356	GOJ 56% Gastric 44%	Median DFS 18 vs. 30 months HR 0.75 (0.62–0.91) *p* = 0.0036	Median OS 38 vs. 50 months HR 0.77 (0.63–0.94) *p* = 0.012
OE02	Preoperative CF vs. Surgery alone	400 402	Oesophageal 25.7% GOJ 64.1% Gastric 10.2%	DFS 26.4% vs. 14.3% HR 0.82 (0.71–0.95) *p* = 0.003	5-year OS 23.0% vs. 17.1% HR 0.84 (0.72–0.98) *p* = 0.03

DFS—disease free survival, OS—overall survival, HR—hazard ratio, CI—confidence Interval, ECF—epirubicin, cisplatin and fluorouracil (5-FU), ECX—epirubicin, cisplatin and capecitabine, GOJ—gastro–oesophageal junction, CF—cisplatin and 5-FU, FLOT—5-FU, leucovorin, oxaliplatin and docetaxel. *Note.* Adapted from “Neoadjuvant and adjuvant multimodality therapies in resectable esophagogastric adenocarcinoma,” by Lau et al. [7]

**Table 2 cancers-14-00586-t002:** Key studies of adjuvant chemotherapy in resectable OG cancers.

Trial	Therapy Arms	N	Tumour Location	DFS, HR (95% CI)	OS, HR (95% CI)
ACTS-GC	S-1 vs. Observation	529 530	All gastric	3-year RFS 72.2% vs. 59.6% HR 0.62 (0.50–0.77) *p* < 0.001	3-year OS 80.1% vs. 70.1% HR 0.68 (0.52–0.87) *p* = 0.003
JACCRO GC-07 (START-2)	Docetaxel + S-1 vs. S-1 alone	454 459	All gastric	3-year RFS 65.9% vs.49.6% HR 0.632 99% CI, 0.400–0.998 *p* < 0.001	NR
CLASSIC	CAPOX vs. Observation	520 515	Gastric 98% GOJ 2%	5-year DFS 68% vs. 53% HR 0.58 (0.47–0.72) *p* < 0.0001	5-year OS 78% vs. 69% HR 0.66 (0.51–0.85) *p* = 0.0015
CheckMate 577 *	Nivolumab vs. placebo	532 262	Oesophageal 60% GOJ 40%	Median DFS 19.4 vs. 11.1 months HR 0.75 (0.59–0.96)	NR

RFS—relapse free survival, S-1—tegafur, gimeracil and oteracil, CAPOX—capecitabine and oxaliplatin, NR—not reported. * Only adenocarcinoma results displayed for the CheckMate 577 trial. *Note.* Adapted from “Neoadjuvant and adjuvant multimodality therapies in resectable esophagogastric adenocarcinoma,” by Lau et al. [7].

**Table 3 cancers-14-00586-t003:** Key studies of adjuvant and neoadjuvant chemoradiotherapy in resectable OG cancers.

Trial	Therapy Arms	N	Tumour Location	DFS, HR (95% CI)	OS, HR (95% CI)
Intergroup-0116	Adjuvant CRT vs. Surgery alone	281 275	GOJ 20% Gastric 80%	Median RFS 30 vs. 19 months HR 1.52 (1.23–1.86) *p* < 0.001	Median OS 36 vs. 27 months HR 1.35 (1.09–1.66), *p* = 0.005
ARTIST	Adjuvant CRT vs. Adjuvant XP	230 228	All gastric	3-year DFS 78.2% vs. 74.2% HR 0.740 (0.52–1.05) *p* = 0.092	5-year OS 75% vs. 73% HR 1.130 (0.78–1.65) *p* = 0.53
CRITICS	Adjuvant ECX vs. Adjuvant CRT (Both after neoadjuvant ECX and surgery)	392 389	GOJ 17% Gastric 83%	Median DFS 28 vs. 25 months HR 0.99 (0.82–1.19) *p* = 0.92	Median OS 43 vs. 37 months HR 1.01 (0.84–1.22) *p* = 0.90
CROSS *	Neoadjuvant CRT vs. Surgery alone	148 141	Distal oesophagus 58% GOJ 24%	Median DFS 29.9 vs. 17.7 months HR 0.69 (0.52–0.92) *p* = 0.010	Median OS 43.2 vs. 27.1 months HR 0.73 (0.55–0.98) *p* = 0.038

CRT—chemoradiotherapy, XP—capecitabine and cisplatin. * Only adenocarcinoma results displayed for the CROSS trial. *Note.* Adapted from “Neoadjuvant and adjuvant multimodality therapies in resectable esophagogastric adenocarcinoma,” by Lau et al. [7].

**Table 4 cancers-14-00586-t004:** A selection of active studies investigating chemoimmunotherapy in resectable OG cancers.

Trial	Phase	N	Tumour Location	Control Arm	Experimental Arm	Stratification Factors	Primary Outcome
Peri-operative							
KEYNOTE 585	III	800	OG/GOJ adenocarcinoma	Chemo + placebo	Chemo + Pembrolizumab	Stage II versus III versus IVa	pCR, OS, EFS
NCT03221426						Asia vs. non-Asia	
						XP/FP versus FLOT	
MATTERHORN	III	900	OG/GOJ adenocarcinoma	FLOT + placebo	FLOT + Durvalumab	PD-L1 TIP > 1% vs. <1%	EFS
NCT04592913						Node status (N+ /N−)	
						Asia vs. non-Asia	
DANTE	II	295	OG/GOJ adenocarcinoma	FLOT	FLOT + Atezolizumab	Node status (N+ /N−)	DFS/PFS
NCT03421288						MSI/dMMR vs. MSS/pMMR	
						Location of primary (GEJ type I vs. GEJ type II/III vs. stomach)	
						*Quantitative PDL-1 mRNA expression will be performed but not a stratification factor*	
NICE	II	80	OG/GOJ adenocarcinoma	XELOX or SOX	XELOX or SOX + SOO1 (PD-1 inhibitor)		Pathological response
NCT04744649							
ICONIC	II	40	OG/GOJ adenocarcinoma		FLOT + Avelumab		pCR
NCT03399071							
NCT04661150	II	52	HER 2 positive GOJ/Gastric adenocarcinoma	CAPOX + Trastuzumab	CAPOX + trastuzumab + Atezolizumab		pCR
NCT04908566	II	124	GOJ/Gastric adenocarcinoma	SOX + PD-1 antibody	FOLFIRINOX + PD-1 antibody		Pathological response
NCT04354662	II	35	OG/GOJ adenocarcinoma		FLOT + Toripalimab		3 y DFS, pCR
NEONIPIGA	II	32	dMMR/MSI OG adenocarcinoma		Ipilimumab + Nivolumab neoadjuvant		pCR
NCT04006262							
Adjuvant							
VESTIGE	II	240	Stage Ib-IVa gastric or GOJ adeno, high risk recurrence (ypN+ +/− R1)	Chemo (as per ESMO guidelines 2016)	Nivolumab/Ipilimumab		DFS
NCT03443856							
NCT03006705	III	700	pStage III OG/GOJ cancer	S-1 or CAPOX + placebo	S-1 or CAPOX + Nivolumab		Relapse free survival

OG—oesophago–gastric, GOJ—gastro–oesophageal junction, FLOT—5-FU, leucovorin, oxaliplatin and docetaxel, CAPOX—capecitabine and oxaliplatin, pCR—pathological complete response, OS—overall survival, EFS—event-free survival, PFS—progression-free survival, DFS—disease-free survival, p—pathological.

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
