# Peer review of "Optimising Multimodality Treatment of Resectable Oesophago-Gastric Adenocarcinoma"

_cancers, 2022, doi:10.3390/cancers14030586_

Round 1

Reviewer 1 Report

This review is very well written. The authors focused on a very hot topic in upper GI cancers and it is definitely an excellent synthesis of current knowledge in this field.

Author Response

Many thanks for your time and kind review of our submission. 

Reviewer 2 Report

The authors reviewed the developments of multimodality strategies in the oesophago-gastric adenocarcinoma. They also outlined ongoing clinical trials in this field and described perspective on biomarker establishment. This manuscript is comprehensive and well-written. However, there are some concerns to be addressed.

  1. Regarding drug names, fluorouracil was used in the table abbreviation, while 5FU was used in the manuscript text. I am wondering if 5FU should be 5-FU. Anyway, this should be unified. In addition, since TS-1 is a product name, ACTS-GC should be Adjuvant Chemotherapy Trial of S-1 for Gastric Cancer (Page 4 Line 150-151).
  2. In Page6 Line 264-269, the results of the Neo-AEGIS trial were reported in not ESMO2021, but ASCO2021.
  3. In the Neo-AEGIS trial, most of enrolled patients did not receive FLOT regimen due to historical backdrop. Therefore, the RACE trial (NCT04375605) is under investigation to evaluate the benefit of adding radiotherapy in patients with GOJ cancers. I would recommend mentioning this.
  4. The authors mentioned the RAMSES trial and the PETRARCA trial. However, the phase III part of the RAMSES trial is ongoing after excluding type I GOJ cancer, while the PETRARCA trial was terminated due to the negative results from the JACOB study. This should be described.
  5. “pP = 0.08” should be “p = 0.08” (Page 9 Line 377).
  6. dMMR seems to be more common than MMRd (Page 9 Line 378).
  7. The authors stated that molecular features of MSI-H or EBV tumors are rationale for potential response of immunotherapy (Page10-11). This is true, but even more important is the fact that several reports have already revealed immunotherapy was effective for these types in clinical setting (e.g. Kim, et al. Nat Med. 2018 24(9): 1449-1458. Mishima, et al. J Immunother Cancer. 2019 7(1): 24.).
  8. “NCT0479566” should be “NCT04795661” (Page 20 Line 396).

Author Response

Thank you for your time and effort in reviewing our manuscript. Please see replies to your points below: 

Point 1: Regarding drug names, fluorouracil was used in the table abbreviation, while 5FU was used in the manuscript text. I am wondering if 5FU should be 5-FU. Anyway, this should be unified. In addition, since TS-1 is a product name, ACTS-GC should be Adjuvant Chemotherapy Trial of S-1 for Gastric Cancer (Page 4 Line 150-151).

Response 1: fuorouracil is now abbreviated as 5-FU and this has been unified in the manuscript text and table abbreviations. S-1 has been corrected (Page 4 Line 150-151)

Point 2: In Page6 Line 264-269, the results of the Neo-AEGIS trial were reported in not ESMO2021, but ASCO2021.

Response 2: This has now been corrected and reference added. In addition, the NCT number was incorrect for this trial and this has been updated to NCT01726452 (Page 6 Line 265). 

Point 3: In the Neo-AEGIS trial, most of enrolled patients did not receive FLOT regimen due to historical backdrop. Therefore, the RACE trial (NCT04375605) is under investigation to evaluate the benefit of adding radiotherapy in patients with GOJ cancers. I would recommend mentioning this.

Response 3: This has now been discussed in Page 7 Line 277-281. 

Point 4: The authors mentioned the RAMSES trial and the PETRARCA trial. However, the phase III part of the RAMSES trial is ongoing after excluding type I GOJ cancer, while the PETRARCA trial was terminated due to the negative results from the JACOB study. This should be described.

Response 4: these points have now been described. JACOB was in the advanced setting and we could not confirm in literature if the PETRARCA trial was close prematurely because of the negative results of JACOB study. 

Point 5: “pP = 0.08” should be “p = 0.08” (Page 9 Line 377).

Response 5: this has now been corrected. 

Point 6: dMMR seems to be more common than MMRd (Page 9 Line 378).

Response 6: this has now been updated to dMMR.

Point 7: The authors stated that molecular features of MSI-H or EBV tumors are rationale for potential response of immunotherapy (Page10-11). This is true, but even more important is the fact that several reports have already revealed immunotherapy was effective for these types in clinical setting (e.g. Kim, et al. Nat Med. 2018 24(9): 1449-1458. Mishima, et al. J Immunother Cancer. 2019 7(1): 24.).

Response 7: these aforementioned reports are based on response to immunotherapy in the advance (metastatic) setting of gastric cancer, which could well be applicable to the early stages setting. Studies are ongoing in the resectable setting. A sentence has been added to explain these reports in the biomarkers section (Page 9 Line 375-378). 

Point 8: “NCT0479566” should be “NCT04795661” (Page 20 Line 396).

Response 8: this has now been corrected. 

Reviewer 3 Report

I read with great pleasure the paper “Optimising multimodality treatment of operable oesophagogastric adenocarcinoma” submitted to Cancers by AA Mohamed.

It is a comprehensive and interesting updated review on multimodality strategies  and biomarkers for the treatment of this disease.

I have some minor comments on this paper:

In my opinion it would be better to consider the term “resectable” instead of “operable oesophagogstric adenocarcinoma”  in the title. 

I would suggest to divide table 1 into 3 tables referring to subparagraphs reported in the text. These paragraphs should also be accordingly re-nominated:

perioperative and neoadjuvant chemotherapy,

neoadjuvant chemoradiotherapy,

adjuvant chemo- and chemioradiotherapy.  

Please check and correct references number 5, 6 and 19.

Author Response

Many thanks for your time and effort in reviewing our manuscript. Please see replies to your points as below: 

Point 1: In my opinion it would be better to consider the term “resectable” instead of “operable oesophagogstric adenocarcinoma”  in the title. 

Response 1: Both terms are used in literature. We agree with you that "resectable" may be more accurate as operability involves other non-surgical aspects such as patient's fitness etc.  This has now been changed to "resectable" as suggested.  

Point 2: I would suggest to divide table 1 into 3 tables referring to subparagraphs reported in the text. These paragraphs should also be accordingly re-nominated: perioperative and neoadjuvant chemotherapy, neoadjuvant chemoradiotherapy, adjuvant chemo- and chemioradiotherapy.  

Response 2: Table 1 has now been re-structured into 3 tables referring to the subparagraphs as suggested. These have been re-named as: perioperative and neoadjuvant chemotherapy, adjuvant chemotherapy, and adjuvant and neoadjuvant chemoradiotherapy. 

Point 3: Please check and correct references number 5, 6 and 19.

Response 3: These references have now been updated to the correct format.